# Scanning Accuracy of Bracket Features and Slot Base Angle in Different Bracket Materials by Four Intraoral Scanners: An In Vitro Study

**DOI:** 10.3390/ma14020365

**Published:** 2021-01-13

**Authors:** Seon-Hee Shin, Hyung-Seog Yu, Jung-Yul Cha, Jae-Sung Kwon, Chung-Ju Hwang

**Affiliations:** 1Department of Orthodontics, Institute of Craniofacial Deformity, Yonsei University College of Dentistry, Seoul 03722, Korea; muze826@naver.com (S.-H.S.); yumichael@yuhs.ac (H.-S.Y.); jungcha@yuhs.ac (J.-Y.C.); 2Department and Research Institute of Dental Biomaterials and Bioengineering, Yonsei University College of Dentistry, Seoul 03722, Korea; jkwon@yuhs.ac; 3BK21 FOUR Project, Yonsei University College of Dentistry, Seoul 03722, Korea

**Keywords:** intraoral scanner, orthodontic bracket, accuracy, precision, trueness

## Abstract

The accurate expression of bracket prescription is important for successful orthodontic treatment. The aim of this study was to evaluate the accuracy of digital scan images of brackets produced by four intraoral scanners (IOSs) when scanning the surface of the dental model attached with different bracket materials. Brackets made from stainless steel, polycrystalline alumina, composite, and composite/stainless steel slot were considered, which have been scanned from four different IOSs (Primescan, Trios, CS3600, and i500). SEM images were used as references. Each bracket axis was set in the reference scan image, and the axis was set identically by superimposing with the IOS image, and then only the brackets were divided and analyzed. One-way analysis of variance (ANOVA) was used to compare the differences. The difference between the manufacturer’s nominal torque and bracket slot base angle was 0.39 in SEM, 1.96 in Primescan, 2.04 in Trios, and 5.21 in CS3600 (*p* < 0.001). The parallelism, which is the difference between the upper and lower angles of the slot wall, was 0.48 in SEM, 7.00 in Primescan, 5.52 in Trios, 6.34 in CS3600, and 23.74 in i500 (*p* < 0.001). This study evaluated the accuracy of the bracket only, and it must be admitted that there is some error in recognizing slots through scanning in general.

## 1. Introduction

Intraoral scanners (IOS) are used in dentistry as a convenient method of taking impressions [1,2,3,4,5,6]. Orthodontic tooth movement can also be easily evaluated using IOS [7,8]. Through the dental model acquired at the time of re-diagnosis of patients undergoing orthodontic treatment, not only the relationship between the entire dentition and the arch but also the position of the bracket is reevaluated. The placing of straight archwire in preadjusted brackets produce three-dimensional tooth-moving forces as a result of the intimate fit of wire into the bracket slot [9,10,11,12]. Therefore, during orthodontic treatment, the position, height, torque, and angulation of the bracket are very important components that have a great influence on the treatment outcome. To evaluate whether the bracket prescription is accurately expressed by checking the height, position, and angle of the bracket slot can help produce a perfect treatment result [13]. After making a model with an impression, there are cases where wiring is done. In this case, since the bracket acts as an undercut of the impression, the impression is usually made with wax attached; the patient’s discomfort is large, and the bracket part is hardly visible. Using an IOS may reduce patient discomfort, and more accurate evaluation may be possible in this area. However, there is no research on whether detailed areas such as the angle or shape of the bracket prescribed as an IOS can be accurately scanned. In addition, products such as Suresmile are provided by bending archwire with a robot with data scanned by IOS during treatment [14]. In this case, the angle and position of the bracket slot must be scanned very accurately to enable wire bending and torque [15], but they are already used under the premise that the bracket scan is accurate. Jung et al. [16] reported that when the bracket-attached model and the bracket-wire ligated model were scanned with IOS, they showed a significant difference in accuracy in the horizontal and vertical measurement items compared to the model without the bracket. Park et al. [17] reported that the horizontal and vertical measurement of the arch with the lingual bracket showed a significant difference in accuracy compared to the arch with the buccal bracket. However, there is no study on whether the angle and shape of the bracket slot are accurately scanned.

In addition, depending on the material properties, there may be differences in the performance of the IOS. In a study on the effect of the material surface on the scan error of IOS, Kurz et al. [18] reported that the error was greater in the resin and metal groups than in the ceramic. Song et al. [19] applied artificial saliva to the maxillary model with non-bracket, ceramic, metal, and resin brackets and scanned them with CS3600, i500, Trios 3, Omnicam IOSs. In this study, the mean and the maximum discrepancy value were evaluated, and it was confirmed that the discrepancy of the dentition with the resin and metal bracket was greater than that of the ceramic bracket. However, because the scan images of the entire maxillary dentition were superimposed to evaluate the discrepancy, the shape of the bracket or the angle of the slot could not be confirmed.

The accuracy of IOS is divided into precision and trueness [20]. The precision refers to the degree to which data acquired by repeating scans under the same conditions match each other. Trueness refers to the ability to reproduce the overall arch close to the real without three-dimensional deformation or distortion.

IOS can be classified into active triangulation, confocal microscopy, optical coherence tomography, and active wavefront sampling according to the data capture principle. Depending on the data capture mode, it can be classified as a system that acquires and stitches individual images or a video sequence system, ultrafast optical sectioning technique. The CS3600 and i500 are scanners using active triangulation. The CS3600 is a video sequence system, and the i500 is a method of stitching images. Trios 3 uses the confocal microscopy principle and ultrafast optical sectioning technique. The recently released Primescan uses a new scanning technique, high-frequency contrast analysis, and dynamic depth scan. In this study, we studied to confirm whether the four principles show differences in accuracy using different scanners.

The aim of this study was to evaluate the accuracy of digital scan images of brackets produced by four IOSs when scanning the surface of the dental model attached with different bracket materials. The null hypothesis was that there were no differences in scanning accuracy depending on the type of scanners and there were no differences in scanning accuracy depending on the materials.

## 2. Materials and Methods

### 2.1. Study Model Design

Two upper dental study models (Dentiform, Tomy Inc., Fuchushi, Japan) were prepared. The horizontal axis was marked using the 019 × 025 stainless steel wire at the position to attach the bracket on the dentiform, and the vertical axis was marked in advance based on the tooth axis. Brackets were bonded on the buccal side with direct passive bracketing using 019 × 025 stainless steel wire from right second premolar to left second premolar. The brackets used in model A were a Bionic metal MBT022 bracket (Ortho Technology, Lutz, FL, USA) for the right teeth and a Reflections ceramic MBT022 bracket (Ortho Technology, Lutz, FL, USA) for the left teeth, and in model B, they were a resin bracket Purfit I resin MBT022 bracket (US Orthodontic products, Norwalk, CA, USA) for the right teeth, and a Purfit II resin MBT022 bracket with metal slot (resinmetal bracket) (US Orthodontic products, Norwalk, CA, USA) for the left teeth. The Purfit II resinmetal bracket had the same design as the Purfit I resin bracket, and only slots are metal (Table 1).

### 2.2. Scanning Electron Microscope (SEM)

All brackets were analyzed by SEM (S-3000N, Hitachi, Tokyo, Japan). The specimens were mounted on SEM studs and dried with a freeze dryer (ES-2030, Hitachi, Tokyo, Japan). Platinum was sputtered to a thickness of 100 nm using an ion coater (E-1010, Hitachi, Tokyo, Japan). Photomicrographs at 20 times magnification were taken from the face and both sides of the bracket at an operating voltage of 15 kV. The torque and bracket slot angle of brackets were measured with a computer-based measuring tool (Image-Pro 10, version 10.0.7, Media Cybernetics, Rockville, MD, USA).

A line coincident with the slot reference line on the bracket was drawn (R) (Figure 1). The corners of the base of the bracket slot were round, so a line parallel to the slot base (dotted line) at a distance of 0.1 mm from the slot base was drawn (B). Similarly, a line parallel to the upper wall of the slot at a distance of 0.1 mm from the upper wall of the slot was drawn (U), and a line parallel to the lower wall of the slot at a distance of 0.1 mm from the lower wall of the slot was drawn (L). The angle formed by lines R and B as the slot base angle (SBA), the angle formed by lines R and U as the upper angle (UA), and the angle formed by lines R and L as the lower angle (LA) were denominated. The difference of SBA was calculated by comparing the SBA measured on each scanned image with the nominal torque provided by manufacturer (Difference of SBA = |nominal torque − SBA|). The absolute value of the difference between the measured UA and LA (ABS angle = |UA − LA|) was calculated to compare the parallelism of the slot wall. All measurements were performed twice at 30 days intervals to ensure the reliability of the studied data.

### 2.3. Scanning Process

This study evaluated 4 types of digital intraoral scanners and 1 extraoral scanner as a reference: Trios 3 (3shape, Copenhagen, Denmark), CS3600 (Carestream Dental LLC, Atlanta, GA, USA), Medit i500 (Medit, Seoul, Republic of Korea), Primescan (Dentsply Sirona, York, PA, USA), and E4 (3shape, Copenhagen, Denmark) (Table 2).

An extraoral scanner, E4 (E4 Dental Scanner; 3Shape, Copenhagen, Denmark), was calibrated in accordance with the manufacturer’s instructions. Study models were digitized as the reference model using the E4 scanner at a constant room temperature (23 °C) following the manufacturers’ instructions. The manufacturer reports the accuracy of this scanner as 4 μm.

Models were scanned by the one operator using 4 intraoral scanners, according to the manufacturer’s recommendation. No powders were applied to the models during scanning. Four intraoral scanners were used to scan parts with the same type of bracket attached. Scanning was started with 2nd premolar and continued to incisor along the occlusion. First, the occlusal surfaces were scanned followed by the lingual and buccal surfaces. When scanning the occlusal surfaces, the scanner head was kept at 0–5 mm from the teeth. For scanning the lingual and buccal surfaces, the scanner tip was rolled 45° to 90° to the lingual and buccal sides, respectively. The image was continuously checked that no areas were missed with the screen.

### 2.4. Datasets

All datasets were converted to STL (stereolithography) files via manufacturers’ certified software for standardization. The parts of each study model with the same type of bracket attached were scanned, 5 times repeatedly (E4, S1–S5). Each study model was scanned 5 times repeatedly by 4 intraoral scanners. (IOS, S1–S5). As a result, 80 IOS datasets were produced in this study.

### 2.5. Scan Data Analysis

#### 2.5.1. Setting the Axis of the Bracket

All scanned data processing was performed using the Geomagic control X program (2020.1.0, 3D systems, Rockhill, SC, USA). The axis of the bracket was set based on the tooth axis in order to compare by separating only the brackets. Each image was trimmed just below the gingival line in order to minimize the data size to facilitate analysis and to exclude artifacts in unimportant areas [21]. Each tooth with a bracket attached was separated. The five scan data (S1–S5) were divided into five teeth: central incisor, lateral incisor, canine, 1st premolar, and 2nd premolar. The y-axis was set tangent to the labial surface of the tooth and to include the bracket base from the sagittal view. The x-axis was set to be perpendicular to the y-axis and parallel to the slot at the face of the bracket. The z-axis was set so that the incisal portion of the bracket wing was bisected at the axial view and the bracket slot base was bisected at the sagittal view and perpendicular to the xy plane on the bracket base. The setting of the axial direction was completed by checking the origin of the bisector of the sagittal, front, and axial direction of the bracket. In this way, the axis of the bracket and single tooth of an E4 S1 scan image were set. The axes were set for each of the 5 separated teeth, and 4 types of bracket materials (metal, ceramic, resin, and resinmetal) were performed in the same way. The remaining 24 scanned images (E4 S2–S5, 4 IOS S1–S5) of the same tooth with the same bracket material were loaded one by one using the E4 S1 image set in the Geomagic control X program as reference data. By using the alignment function between the measured data, the entire optimal alignment was performed, and the axis was set equally based on the teeth. The base plane was set as the xy plane, and only the bracket was uniformly divided by the base plane with the z-axis as the normal direction from the origin (Figure 2). Comparison of images of devided brackets was shown Figure 3 and Figure 4.

#### 2.5.2. Bracket Slot Angle

Section 1 was formed by cutting the bracket with the yz-plane (Figure 5). Section 2 and Section 3 were formed by cutting the brackets at a distance of −1 mm and +1 mm to the x-axis from the yz plane. Similar to Figure 1, in each section, a line was drawn tangent to the bracket slot base (B), and its contour was defined as the intersection of two points marked on this wall at a distance of 0.1 mm from the bracket slot wall. Two lines was drawn tangent to the upper wall of the bracket slot (U) and tangent to the lower wall of the bracket slot (L) at a distance of 0.1 mm apart from the slot base and slot face. Depending on the scanner type, when the line angle was more rounded, a line was drawn tangent only with a straight slot wall excluding the round part. On each section, B, U, and L lines were drawn in the same way as SEM measurement (Figure 1), and the slot base angle (SBA), upper angle (UA), and lower angle (LA) for the xy plane were measured. The average of the three values measured in Sections 1, 2, and 3 was calculated. Difference of the SBA (=|nominal torque − SBA|) and ABS angle (=|UA − LA|) were calculated.

#### 2.5.3. Superimposition

The precision of the data repeatedly measured 5 times in each IOS was measured by cross-comparison for each bracket. Within each IOS, two brackets were superimposed based on the axis of the bracket. The error between the two brackets in all data point clouds was calculated as the RMS value.
(1)$$\mathrm{RMS}=\frac{1}{\sqrt{n}}.\sqrt{\sum_{i=1}^{n}{\left(X_{1,i}-X_{2,i}\right)}^{2}}$$X_1,i_ and X_2,i_ is the i point of two brackets of IOS, and n is the number of all measured points. The RMS value shows how far the deviation is from zero between the other two values of the data. So, when the RMS value is low, it indicates that the overlapped data is three-dimensionally consistent. The color range was set to 0–0.2 mm, and the result was displayed as a colormap. Primescan, Trios3, CS3600, i500 were all performed.

### 2.6. Statistical Analysis

All statistical analyses were performed using IBM SPSS software, version 25.0 (IBM Korea Inc., Seoul, Korea) for Windows. The mean, standard deviation (SD), median, and quartile were used to describe the distribution of each variable in this study. When N ≤ 30, the Kolmogorov–Smirnov test and Shapiro–Wilk test were carried out to verify the normality of each variable. For total precision, one-way analysis of variance (ANOVA) and post-hoc Tukey test were used. The differences of SBA for each tooth were tested by the Kruskal–Wallis test, and those for all teeth were tested by ANOVA and a post hoc Tukey test. ABS angles were tested by Kruskal-Wallis test and Mann-Whitney U test post-hoc pairwise comparisons. *p* Values less than 0.05 were considered statistically significant.

## 3. Results

### 3.1. Precision

The precision was shown in Table 3 and Figure 6. As a result of the post-hoc Tukey test, significant differences between IOSs in the same bracket were shown in uppercase letters. In all brackets, the precision was significantly different in the order of Trios 3 < Primescan < CS3600 < i500 (*p* < 0.001). In Primescan and Trios 3, RMS values were small for metal and ceramic brackets, and they were significantly larger for resin and resinmetal brackets (*p* < 0.05).

### 3.2. Trueness

#### 3.2.1. Difference of SBA

The SBA differences calculated for each tooth are shown in Table 4. i500 was excluded because the slot base was round (Figure 3 and Figure 4). There was no significant difference between teeth and brackets, and there was a significant difference between scanners in the same tooth. The same trend was observed between scanners in the same bracket. The difference of SBA for all teeth was 0.39 ± 0.31 in SEM, 1.96 ± 0.16 in Primescan, 2.04 ± 1.95 in Trios 3, and 5.21 ± 4.32 in CS3600 (*p* < 0.001). There was no significant difference between Primescan and Trios 3, and there were significant differences in all others.

#### 3.2.2. ABS Angle

The upper angle and lower angle of the bracket slot are shown in Figure 7. The lower angle had less error than the upper angle. The ABS angle was calculated as the absolute value of the difference between the upper and lower angle and showed the parallelism of the slot. Calculated discrepancies were compared with those measured in the SEM (Table 5). The mean of the ABS angle was 0.48 ± 0.29 in SEM, 7.00 ± 7.08 in Primescan, 5.52 ± 5.37 in Trios 3, 6.34 ± 5.40 in CS3600, and 23.74 ± 10.02 in i500 (*p* < 0.001). In other words, the parallelism of the bracket slot wall was not significantly different between Primescan, and Trios 3, CS3600. They had significantly greater difference than SEM, and i500 was significantly greater than them. There was no significant difference for error according to the bracket materials.

## 4. Discussion

This study evaluated the performance of four types of IOS by limiting it to the bracket scanning images. There were significant differences in the scanning accuracy of the four different bracket materials, and there were significant differences in the scanning accuracy of the four IOSs. Therefore, both null hypotheses were rejected. The RMS value of the precision was small in the metal bracket and ceramic bracket, and it was large in the resin and resinmetal brackets. The translucency of the material may have contributed to this result given the effect of the bracket within the same scanner. Kurtz et al. [18] said that when scanning with the triangulation principle IOS, the discrepancy in the metal and the resin material was higher, but the scan noise according to the material type was within the range of the measurement error existing in the conventional impression, so it could be clinically acceptable. However, especially in the presence of water, the error was much larger than the measurement error according to the material, and clinically relevant errors occurred. This was because the deviation of the angle measurement increased due to the refraction of light in water during scanning [18]. Li et al. [22] reported that objects with higher translucency objects resulted in lower scanning accuracy and larger morphological changes when scanned with IOS using the confocal microscopy principle. The ceramic bracket was more accurate than the resin, which seems to be because the polycrystalline ceramic bracket used in this study has less light reflectivity. Song et al. [19] stated that the largest discrepancy was in the order of resin > metal > ceramic bracket. In this study, it is thought that the discrepancy of the metal bracket was relatively small because the size of the bracket was smaller than that of the resin or ceramic bracket.

The bracket slot angle was divided into SBA (slot base angle) and wall parallelism. For bracket angle measurements (torque and parallelism) according to ISO 27020:2010 [23], a manufacturing error of ±1 is permitted. Regarding the parallelism of the inner wall of the slot, Major et al. [24] measured the manufacturing tolerance of the orthodontic bracket slot and reported a convergent taper of 1.47°, a slightly divergent taper below that, and the most rectangle in shape depending on the product. Araujo et al. [25] reported that there was no significant difference in torque manufacturing tolerance, and that it converged with respect to the parallelism of the inner wall of the bracket slot, and that the average parallelism was measured from +0.19 to −4.10 depending on the manufacturer [11,26,27]. In this study, the slot base angle was the same as the torque of the bracket prescription, and in order to check the tolerance according to the manufacturer, the torque and parallelism of the actual bracket were measured with a scanning electron microscope, and it was confirmed that it was within the manufacturing error.

When comparing the absolute difference between the nominal torque and the measured SBA, there was a significant difference between scanners. The mean of the differences was 0.39 in SEM, 1.96 in Primescan, 2.04 in Trios 3, and 5.21 in CS3600. Since there may be differences due to the SEM image and the baseline of the IOS, it is desirable to focus on the trend rather than the numerical value. The mean difference of bracket slot wall was 0.48 in SEM, 7.00 in Primescan, 5.52 in Trios 3, 6.34 in CS3600, and 23.74 in i500. The difference in parallelism of the slot wall is large in IOS compared to SEM. There was no significant difference between Primescan, Trios 3, and CS3600, and Primescan and Trios 3 tended to converge, while CS3600 and i500 tended to diverge. Compared with SEM, the mean discrepancy of the upper angle was larger than that of the lower angle. The difference of SBA and ABS angle did not show a tendency according to the bracket material. The deviation of the digital scan is smallest when the IOS camera is positioned perpendicular to the surface to be scanned and the light is reflected at 90 degrees, and the magnitude of the deviation increases as the camera moves away from the vertical plane [18]. Therefore, the reason why the parallelism of the slot wall is more inaccurate than the slot base angle is thought to be that the light is farther away from the vertical plane of the camera from the wall than the slot base. In addition, the scan noise of the material caused by the reflection or absorption of light exists within the measurement error range. However, it seems that the error due to the bracket material was not confirmed at the slot angle because a larger error occurs in areas where light does not reach well during digital scanning. In addition, there was a difference in the roundness of the line angle of the inner surface where the wall of the slot and the base meet according to the type of scanner (Figure 3 and Figure 4).

In the case of a bracket that is much smaller than that of an intraoral prepared tooth or implant scan body, the difference in scanning accuracy for errors in slot base or line angle seems to be due to the difference in the principle of the IOS [28,29,30]. The CS3600 and i500 use the principle of triangulation. The CS3600, a video sequence system, scanned the bracket a little more accurately than the i500, which stitches images [31,32]. In addition, Trios 3, which uses the principle of confocal microscopy and ultrafast optical sectioning technique, had higher bracket scanning accuracy than these. It is thought that this was because the depth of field was well expressed by the vibration, so that small structures could be accurately scanned. Primescan also showed high accuracy, and the manufacturer describes that Primescan uses high-frequency contrast analysis and dynamic depth scan as a new method of scanning principle. However, little is known about the various scanning strategies, as this aspect has not been clearly explained [33,34,35].

This study has several limitations. Since this study is an in vitro study, it does not reflect the conditions with moisture or scan restrictions. Previous studies have shown that the presence of moisture in oral scanners affects the accuracy of the scanner [36]. Previous studies showed that the accuracy of the IOS was affected by the user’s experience and skill level [37,38]. To minimize the impact of this variation, one researcher performed all scans after sufficiently practicing the use of each IOS, and data processing was also performed by same researcher. This study is a segmentation study that scans from the unilateral central incisor to the second premolar using the maxillary model, and there may be differences in the case of continuous arch. In addition, since the brackets used in this study do not represent many types of brackets, they need to be extended to more types and numbers of brackets. Even if the IOS is the same product, the accuracy may vary depending on the software version. For the i500, this experiment was performed in version 2.2 and the latest software version is currently released up to 2.3.4. According to the manufacturer, there was a firmware upgrade that speeds up scanning and improves shooting capability in metal, so the product using the new version may show different results from this study. Therefore, further research is needed, taking this into account.

## 5. Conclusions

This study evaluated the accuracy of the bracket only. According to the results of this study, it was possible to confirm the bracket slot base angle, which is difficult to obtain by the conventional impression method according to the scanner type. However, it must be admitted that there is some error in recognizing slots through scanning in general. Considering only the scan of the bracket in this study, Primescan and Trios 3 were more accurate among the four types of IOSs: Primescan, Trios 3, CS3600, and i500. Among the brackets, it should be noted that the polycrystalline ceramic bracket, which has less reflection or absorption of light when using the scan, has high precision, and there is more error when using other types of brackets.

## Figures and Tables

**Figure 1 materials-14-00365-f001:**
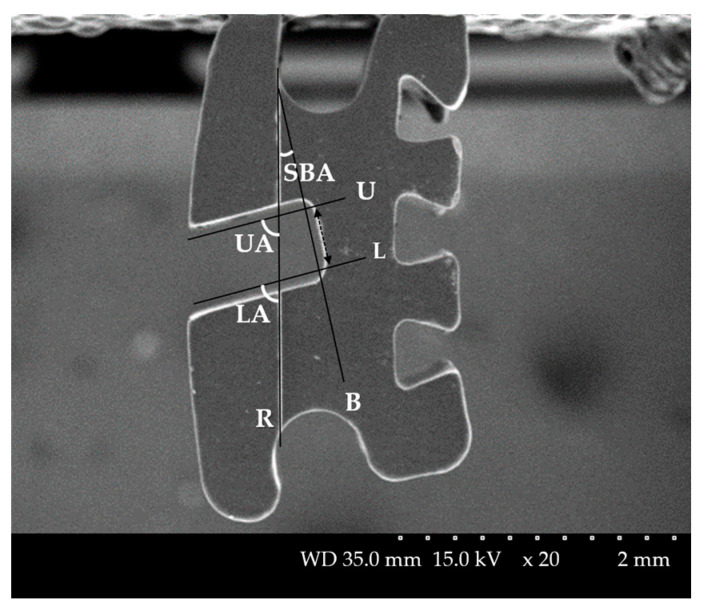
Bracket angles of SEM. Reference line on the bracket: R, A line parallel to the slot base (dotted line) at a distance of 0. 1 mm from the slot base: B, A line parallel to the upper wall of the slot at a distance of 0.1 mm from the upper wall of the slot: U, A line parallel to the lower wall of the slot at a distance of 0.1 mm from the lower wall of the slot: L, The angle between R and B: Slot base angle (SBA), The angle between R and U: Upper angle (UA), The angle between R and L: Lower angle (LA).

**Figure 2 materials-14-00365-f002:**
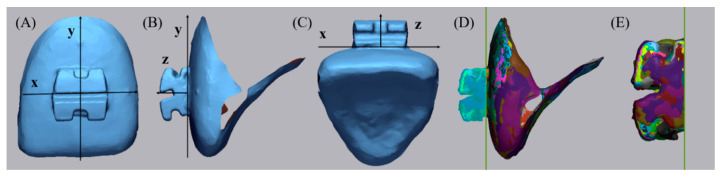
Setting the axis of the bracket: (**A**–**C**) To include the bracket base and tangent to the labial surface of the tooth from the sagittal view: y-axis, to be perpendicular to the y-axis and parallel to the slot at the face of the bracket: x-axis, to be perpendicular to the xy-axis on the bracket base and bisector of the slot base (sagittal view) and incisal portion of bracket wings (axial view): z-axis. (**D**) All aligned ceramic brackets on central incisor: various colors. Only the bracket was uniformly divided by the xy-plane with the z-axis as the normal direction from the origin. (**E**) Divided ceramic brackets.

**Figure 3 materials-14-00365-f003:**
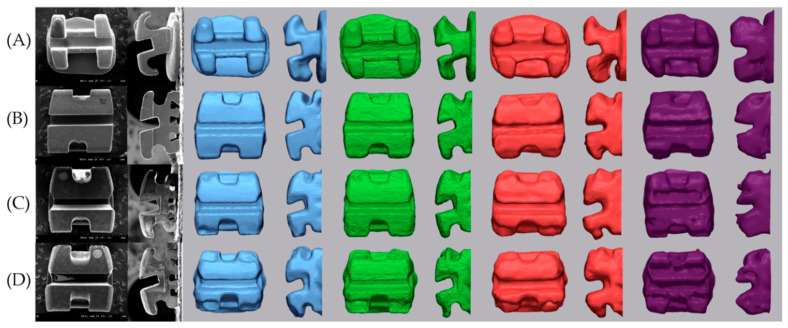
Comparison of images of devided brackets of central incisor: From left to right; SEM micrographs, Primescan, Trios 3, CS3600, i500: (**A**) Metal bracket, (**B**) Ceramic bracket, (**C**) Resin bracket, (**D**) Resin bracket with metal slot.

**Figure 4 materials-14-00365-f004:**
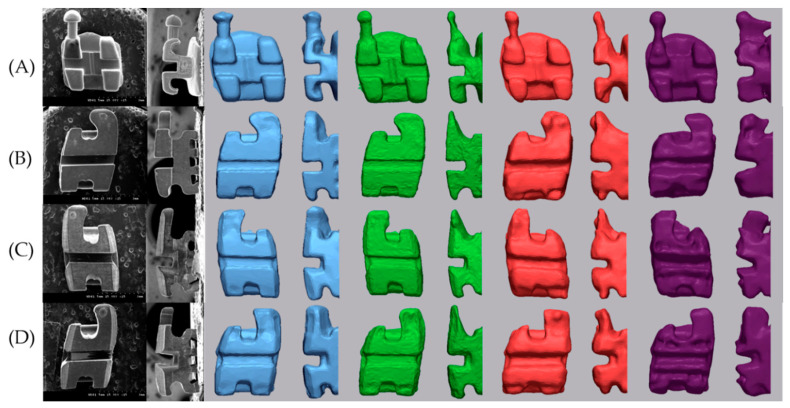
Comparison of images of divided brackets of canine: From left to right; SEM micrographs, Primescan, Trios 3, CS3600, i500 (**A**) Metal bracket, (**B**) Ceramic bracket, (**C**) Resin bracket, (**D**) Resin bracket with metal slot.

**Figure 5 materials-14-00365-f005:**
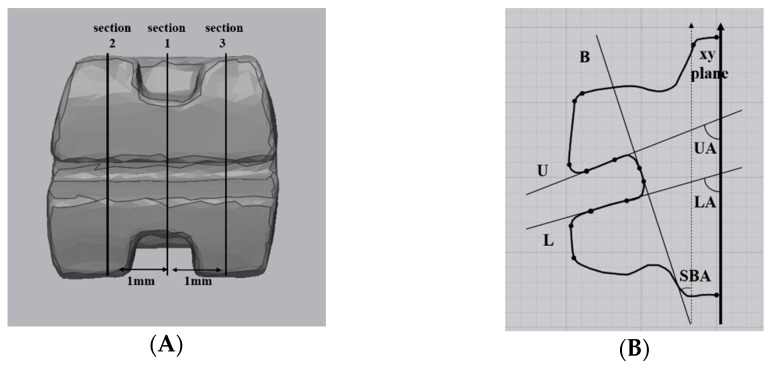
Bracket angles of intraoral scanners (IOS): (**A**) Sections 1, 2, and 3 were formed by cutting the bracket with the yz-plane. (**B**) Tangent to the bracket slot base: B, Tangent to the upper wall of the bracket slot: U, Tangent to the lower wall of the bracket slot: L, The angle between R and B: SBA, The angle between R and U: UA, The angle between R and L: LA.

**Figure 6 materials-14-00365-f006:**
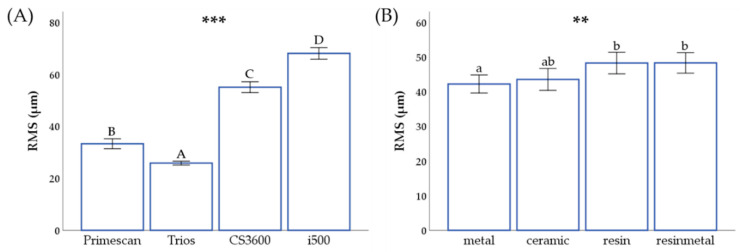
(**A**) Precision RMS between scanners. A,B,C,D Uppercase letters within the same row indicate significant differences between scanners. (**B**) Precision RMS between brackets. Different letters above the bars indicate significant differences. ** *p* < 0.01, *** *p* < 0.001 for comparison between the groups. a,b Lowercase letters within the same column indicate significant differences between brackets.

**Figure 7 materials-14-00365-f007:**
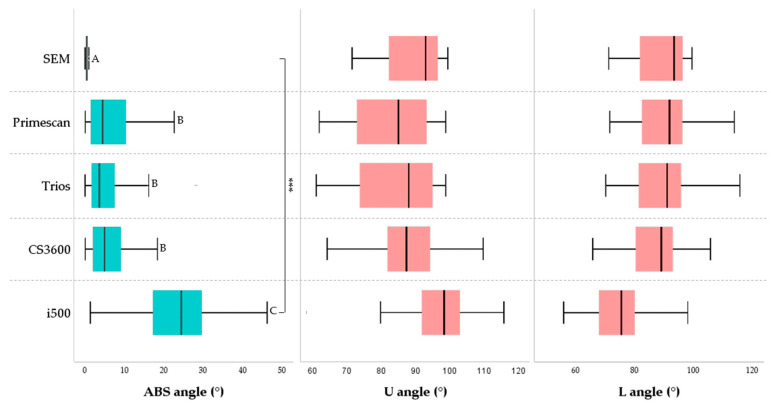
Parallelism of the bracket slot wall. ABS angle represents the parallelism between the upper and lower walls of the bracket slot. A, B, C, uppercase letters beside the bars indicate significant differences between scanners. *** *p* < 0.001 for comparison between the groups.

**Table 1 materials-14-00365-t001:** Summary of the brackets used for experiments.

Material	Model Name	Manufacturer	Dimension (Inches)	Position	Prescription; Torque (°)
Stainless steel	Bionic metal	Ortho Technology, Lutz, FL, USA	022	Model A #11–15	MBT Central incisor; +17° Lateral incisor; +10° Canine; 0° or −7° (Metal, Ceramic; 0°, Resin, Resinmetal; −7°) Premolar; −7°
Polycrystalline alumina	Reflection ceramic	Ortho Technology, Lutz, FL, USA	022	Model A #21–25	
Composite	Purfit I resin	US Orthodontic products, Norwalk, CA, USA	022	Model B #11–15	
Composite, stainless steel slot	Purfit II resin (resinmetal); same design as the Purfit I resin bracket, and only slots are metal	US Orthodontic products, Norwalk, CA, USA	022	Model B #21–25	

**Table 2 materials-14-00365-t002:** Intraoral scanners, manufacturer, scanner technology, light source, acquisition method, and software version.

Product	Manufacturer	Scanner Technology	Light Source	Acqusition Method	Software Version
Primescan	Dentsply Sirona, York, PA, USA	Confocal microscopy	Light	Close to Individual image	5.1
Trios 3	3shape, Copenhagen, Denmark	Confocal microscopy	Light	Ultrafast optical sectioning technique	1.4.7.3
CS3600	Carestream, Rochester, NY, USA	Triangulation	Light	Video sequence	3.1.0
i500	Medit, Seoul, Republic of Korea	Trianulation	Light	Individual image	2.2.4.7

**Table 3 materials-14-00365-t003:** Precision RMS results from superimposition of IOS scanned images with context interaction in all teeth.

RMS (Mean ± SD, μm)						
Bracket Material	Primescan	Trios 3	CS3600	i500	Total	*p*
Metal	27.15 ± 9.17 ^aA^	29.08 ± 4.46 ^bA^	53.24 ± 17.62 ^B^	59.57 ± 12.36 ^aC^	42.26 ± 18.67 ^a^	***
Ceramic	26.88 ± 10.23 ^aA^	24.28 ± 4.24 ^aA^	54.50 ± 13.30 ^B^	68.65 ± 18.70 ^bC^	43.58 ± 22.62 ^ab^	***
Resin	39.53 ± 10.80 ^bB^	22.79 ± 4.84 ^aA^	56.95 ± 14.87 ^C^	73.93 ± 13.15 ^bD^	48.30 ± 22.32 ^b^	***
Resinmetal	39.81 ± 16.82 ^bB^	27.39 ± 5.30 ^bA^	55.75 ± 13.85 ^C^	70.43 ± 15.59 ^bD^	48.34 ± 21.18 ^b^	***
Total	33.34 ± 13.70 ^B^	25.89 ± 5.31 ^A^	55.11 ± 14.95 ^C^	68.14 ± 15.95 ^D^		***
*p*	***	***	NS	***	**	

* *p* < 0.05, ** *p* < 0.01, *** *p* < 0.001, NS: not significant, *p* value calculated one-way ANOVA and Tukey’s post hoc analysis at α = 0.05. ^A,B,C,D^ Uppercase letters within the same row indicate significant differences between scanners. ^a,b^ Lowercase letters within the same column indicate significant differences between brackets.

**Table 4 materials-14-00365-t004:** Difference of SBA for each tooth according to the IOS and bracket materials.

Difference of SBA, Median (Q1–Q3), (°)							
	Bracket	SEM	Primescan	Trios 3	CS3600	*p*	IOS
Central incisor	Metal	0.46 (0.08–0.84)	1.00 (0.64–1.81)	2.73 (1.70–3.70)	1.67 (1.12–3.08)	*	*p* < 0.001
	Ceramic	0.31 (0.19–0.72)	0.69 (0.41–0.82)	3.77 (3.37–5.26)	10.97 (6.00–16.69)	*	
	Resin	0.59 (0.18–0.80)	1.30 (0.70–3.80)	0.81 (0.68–2.35)	5.76 (2.45–7.27)	*	
	Resinmetal	0.35 (0.12–0.80)	3.41 (2.31–3.55)	1.04 (0.64–2.53)	3.61 (3.30–4.92)	*	
	Total	0.45 ± 0.33 ^A^	1.64 ± 1.19 ^B^	2.57 ± 1.54 ^B^	4.87 ± 4.54 ^C^		
	*p*	NS	*	*	*		
Lateral incisor	Metal	0.44 (0.29–0.80)	6.31 (4.85–7.28)	6.11 (4.80–8.66)	8.28 (4.98–10.68)	*	*p* < 0.001
	Ceramic	0.20 (0.04–0.35)	1.29 (0.82–1.66)	0.69 (0.45–3.47)	2.93 (0.86–3.79)	NS	
	Resin	0.35 (0.11–0.62)	1.45 (0.29–2.12)	0.48 (0.23–0.99)	1.47 (0.53–5.47)	NS	
	Resinmetal	0.92 (0.59–1.64)	1.46 (0.44–5.97)	2.88 (2.62–3.74)	1.85 (0.72–3.82)	NS	
	Total	0.53 ± 0.46 ^A^	3.53 ± 2.77 ^B^	3.74 ± 2.99 ^B^	4.03 ± 0.81 ^C^		
	*p*	NS	*	**	*		
Canine	Metal	0.39 (0.32–0.47)	2.11 (1.04–3.43)	1.18 (0.53–2.88)	4.06 (1.50–6.48)	*	*p* < 0.001
	Ceramic	0.16 (0.03–0.32)	3.84 (1.64–4.87)	1.69 (1.10–3.59)	12.76 (4.60–17.79)	*	
	Resin	0.38 (0.06–0.70)	1.54 (0.53–2.67)	0.71 (0.06–1.33)	4.67 (3.17–5.95)	*	
	Resinmetal	0.28 (0.13–0.75)	2.09 (0.74–4.74)	2.83 (1.50–3.69)	6.18 (2.07–6.91)	*	
	Total	0.33 ± 0.25 ^A^	2.56 ± 1.50 ^B^	1.74 ± 1.33 ^B^	5.85 ± 4.55 ^C^		
	*p*	NS	NS	NS	NS		
1st Premolar	Metal	0.35 (0.24–0.45)	0.90 (0.82–1.11)	1.45 (1.11–2.04)	4.49 (2.41–5.63)	***	*p* < 0.001
	Ceramic	0.14 (0.05–0.17)	0.54 (0.19–1.06)	0.66 (0.30–0.92)	6.16 (5.80–10.10)	*	
	Resin	0.44 (0.29–0.63)	0.53 (0.34–0.68)	0.39 (0.20–0.61)	1.47 (0.78–4.76)	*	
	Resinmetal	0.29 (0.14–0.56)	0.58 (0.20–1.30)	0.59 (0.52–1.27)	8.00 (4.58–11.54)	*	
	Total	0.31 ± 0.19 ^A^	0.74 ± 0.40 ^B^	1.01 ± 0.68 ^B^	5.23 ± 3.40 ^C^	*	
	*p*	NS	NS	*	*		
2nd Premolar	Metal	0.31 (0.10–0.77)	2.61 (1.47–2.89)	1.80 (1.03–2.30)	2.80 (0.78–4.52)	*	*p* < 0.001
	Ceramic	0.32 (0.13–0.43)	0.77 (0.51–1.22)	0.40 (0.19–0.83)	6.48 (6.36–9.46)	*	
	Resin	0.27 (0.18–0.48)	1.02 (0.12–1.39)	0.47 (0.21–1.05)	5.39 (1.48–7.68)	*	
	Resinmetal	0.28 (0.13–0.75)	1.47 (0.50–2.85)	1.35 (0.80–1.79)	5.55 (3.02–16.01)	*	
	Total	0.35 ± 0.25 ^A^	1.55 ± 1.03 ^B^	1.15 ± 0.84 ^B^	5.40 ± 5.16 ^C^		
	*p*	NS	*	*	NS		
All teeth		0.39 ± 0.31 ^A^	1.96 ± 0.16 ^B^	2.04 ± 1.95 ^B^	5.21 ± 4.32 ^C^		

* *p* < 0.05, ** *p* < 0.01, *** *p* < 0.001, NS: not significant, *p* value for ‘total’ row of each tooth calculated with Kruskal–Wallis test. ^A,B,C^ Uppercase letters within the same row indicate significant differences between scanners by Mann–Whitney U test post hoc pairwise comparisons at α = 0.0083. *p* Value for all teeth calculated with one-way ANOVA. ^A,B,C^ Uppercase letters within the same row indicate significant differences between scanners by Tukey’s post hoc analysis at α = 0.05.

**Table 5 materials-14-00365-t005:** Mean ABS angle and its standard deviation according to the IOSs and bracket materials.

ABS Angle (Mean ± SD, °)							
Bracket Material	SEM	Primescan	Trios 3	CS3600	i500	Total	*p*
Metal	0.44 ± 0.25 ^A^	2.99 ± 3.76 ^aB^	3.46 ± 3.39 ^aB^	5.60 ± 4.79 ^B^	18.60 ± 8.81 ^aC^	7.01 ± 8.36 ^a^	***
Ceramic	0.52 ± 0.25 ^A^	6.39 ± 4.71 ^bB^	5.94 ± 5.79 ^abB^	6.62 ± 5.38 ^B^	30.11 ± 8.51 ^bC^	10.31 ± 11.84 ^b^	***
Resin	0.34 ± 0.17 ^A^	5.76 ± 6.08 ^a^^bB^	6.07 ± 5.48 ^abB^	5.30 ± 5.72 ^B^	27.63 ± 11.62 ^bC^	9.37 ± 11.82 ^ab^	***
Resinmetal	0.62 ± 0.37 ^A^	16.95 ± 5.70 ^cC^	8.65 ± 6.50 ^bB^	8.59 ± 5.88 ^B^	23.76 ± 6.29 ^abD^	12.18 ± 9.53 ^b^	***
Total	0.48 ± 0.29 ^A^	7.00 ± 7.08 ^B^	5.52 ± 5.37 ^B^	6.34 ± 5.40 ^B^	23.74 ± 10.02 ^C^		***
*p*	NS	***	**	NS	***	***	

* *p* < 0.05, ** *p* < 0.01, *** *p* < 0.001, NS: not significant, *p* value for ‘total’ row and column calculated with one-way ANOVA. ^A,B,C^ Uppercase letters within the same row indicate significant differences between scanners, and ^a,b,c,d^ lowercase letters within the same column indicate significant differences between brackets by Tukey’s post hoc analysis at α = 0.05. *p* value for the others except ‘total’ row and column calculated with Kruskal–Wallis test. ^A,B,C,D^ Uppercase letters within the same row indicate significant differences between scanners by Mann–Whitney U test post hoc pairwise comparisons at α = 0.005, ^a,b,c,d^ lowercase letters within the same column indicate significant differences between brackets by Mann–Whitney U test post hoc pairwise comparisons at α = 0.0083.

## Data Availability

Data sharing is not applicable to this article.

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
