# Peer review of "Scanning Accuracy of Bracket Features and Slot Base Angle in Different Bracket Materials by Four Intraoral Scanners: An In Vitro Study"

_materials, 2021, doi:10.3390/ma14020365_

Round 1

Reviewer 1 Report

read with great interest the Manuscript titled “Scanning Accuracy of Bracket Features and Slot Base Angle in Different Bracket Materials by Four Intraoral Scanners” (ID: materials-1050483), which falls within the aim of Materials. In my opinion, the topic is attracting and of current interest; the text is well written, methodology is accurate and conclusions are supported by the data analysis. Nevertheless, Authors should consider the following recommendations:

  • Please, define the type of statistical analysis in the Abstract section;
  • specify the null hypothesis in the Introduction, rejecting or accepting it in the Discussion section;
  • the study subject is very technical and aimed to an expert audience. Anyway, the authors should try to clarify and emphasize the clinical implications of their study;
  • the authors have correctly cited the limitations of their study; however, they should also highlight the strengths and novelty of their investigation.

Author Response

Comment 1

read with great interest the Manuscript titled “Scanning Accuracy of Bracket Features and Slot Base Angle in Different Bracket Materials by Four Intraoral Scanners” (ID: materials-1050483), which falls within the aim of Materials. In my opinion, the topic is attracting and of current interest; the text is well written, methodology is accurate and conclusions are supported by the data analysis.

Response to Comment 1

Thank you for your encouraging comments and appreciation for the research work.

Comment 2

Please, define the type of statistical analysis in the Abstract section;

Response to Comment 2

Thank you for your valuable input. Following the suggestion, the type of statistical analysis methods are now defined in the Abstract.

Comment 3

specify the null hypothesis in the Introduction, rejecting or accepting it in the Discussion section;

Response to Comment 3

As per the suggestions, the introduction section has been modified to include specifying the null hypothesis, and the null hypothesis was rejected in the discussion section that is now included in manuscript.

Comment 4

the study subject is very technical and aimed to an expert audience. Anyway, the authors should try to clarify and emphasize the clinical implications of their study

Response to Comment 4

The introduction section has been amended as per the thoughtful suggestions, to clarify and emphasize the clinical implication of our investigation.

Comment 5

the authors have correctly cited the limitations of their study; however, they should also highlight the strengths and novelty of their investigation.

Response to Comment 5

Following the suggestions, the introduction section and the end of the discussion section has been amended to highlight strengths and novelty of our study.

Reviewer 2 Report

The topic covered in this study was presented in the title and in the abstract quite interesting and current.

The study presented was an vitro study. I think it's appropriate to insert "an in vitro study" already in the title and therefore clearly in the abstract.

This was the aim of the study: "...to evaluate the accuracy of digital scan images of brackets produced by four different intraoral scanners (IOSs) in terms of ...". Nevertheless the conclusions make clear unequivocally that the study was only able to assess the accuracy of the brackets and not the accuracy of the scanners in detecting the images. Given this, the work should be completely revised and attention should be focused more on brackets and not on scanners.

The limits of this study are numerous and should be extended to a much larger number of brackets (not just two manufacturers).

The study has been described with extreme precision and attention to the smallest detail, producing interesting results. However, we don't fully understand the clinical and practical usefulness of this work.

Surely this paper is able to lay the foundations for the development of numerous other studies on the topic.

Author Response

Comment 1

The study presented was an vitro study. I think it's appropriate to insert "an in vitro study" already in the title and therefore clearly in the abstract.

Response to Comment 1

Thank you for the valuable suggestion, “an in vitro study” has been inserted in the title.

Comment 2

This was the aim of the study: "...to evaluate the accuracy of digital scan images of brackets produced by four different intraoral scanners (IOSs) in terms of ...". Nevertheless the conclusions make clear unequivocally that the study was only able to assess the accuracy of the brackets and not the accuracy of the scanners in detecting the images. Given this, the work should be completely revised and attention should be focused more on brackets and not on scanners.

Response to Comment 2

The authors would like to thank the reviewer for the valuable and helpful comments for improvement of the manuscript. We would like to apologize for misrepresenting the purpose of the study only in the abstract section. The aim in the Abstract has been amended to emphasize more on brackets and not on scanners

Comment 3

The limits of this study are numerous and should be extended to a much larger number of brackets (not just two manufacturers).

Response to Comment 3

Following the thoughtful suggestions, the discussion section has been modified to include the limitations of the study regarding the types and numbers of brackets.

Comment 4

The study has been described with extreme precision and attention to the smallest detail, producing interesting results. However, we don't fully understand the clinical and practical usefulness of this work.

Response to Comment 4

Sorry for the lack of clarity in terms of the clinical and practical usefulness of this work. The introduction section has been amended as per the thoughtful suggestions, to clarify and emphasize the clinical implication of our investigation. Also, the introduction section and the end of the discussion section has been amended to highlight strengths and novelty of our study.

Reviewer 3 Report

Dear authors,

In general, the work is well explained. The design of the study is correct and the results are clearly presented.

Regarding my review I consider that the article can be published in your journal if the authors make some minor changes. The authors describe an important and clinically significant issue in orthodontic materials.

Authors need to review the regulations on how they should write the bibliographic references. Some bibliographical references are not adequately described in the manuscript. This point is important.

Furthermore, I believe that the conclusions of the work should be reformulated to adapt them to the specific objectives described by the authors in the manuscript.

Best regards,

Author Response

Comment 1

In general, the work is well explained. The design of the study is correct and the results are clearly presented. Regarding my review I consider that the article can be published in your journal if the authors make some minor changes. The authors describe an important and clinically significant issue in orthodontic materials.

Response to Comment 1

Thank you for the encouraging comment. The suggested changes for the improvement of the manuscript have been duly noted and rectified.

Comment 2

Authors need to review the regulations on how they should write the bibliographic references. Some bibliographical references are not adequately described in the manuscript. This point is important.

Response to Comment 2

The authors would like to thank the reviewer and apologize for the lapse in formatting. The manuscript has been verified to reflect the changes.

Comment 3

Furthermore, I believe that the conclusions of the work should be reformulated to adapt them to the specific objectives described by the authors in the manuscript.

Response to Comment 3

We would like to apologize for misrepresenting the purpose of the study only in the abstract section. The aim in the Abstract has been amended.